

# Extent, duration and timing of the sea ice cover in Hornsund, Svalbard in 2014-2023

Zuzanna M. Swirad[1], A. Malin Johansson[2], Eirik Malnes[3]

[1]Institute of Geophysics, Polish Academy of Sciences, Warsaw, Poland
[2]UiT The Arctic University of Norway, Tromsø, Norway
[3]NORCE Norwegian Research Centre AS, Oslo, Norway

*Correspondence to*: Zuzanna M. Swirad (zswirad@igf.edu.pl)

**Abstract.** The Sentinel-1A/B synthetic aperture radar (SAR) imagery archive between 14 October 2014 and 29 June 2023 was used in combination with a segmentation algorithm to create a series of binary ice/open water maps of Hornsund fjord,

Svalbard at 50 m resolution for nine seasons (2014/15 to 2022/23). The near-daily (1.57 day mean temporal resolution) maps were used to calculate sea ice coverage for the entire fjord and its parts: the main basin and three major bays: Burgerbukta, Brepollen and Samarinvågen. The average length of the sea ice season was 158 days (range: 105-246 days). Drift ice first arrived from the south-west between October and March and the fast ice onset was on average 24 days later. The fast ice typically disappeared in June, around 20 days after the last day with drift ice. The average sea ice coverage over the sea ice

season was 41% (range: 23-56%), but it was lower in the main basin (27%) compared to the bays (63%). Of the bays, Samarinvågen had the highest sea ice coverage (69%) likely due to the location in southern Hornsund protected from the incoming wind-generated waves and a narrow opening. Seasonally, the highest sea ice coverage was observed in April for the entire fjord and the bays, and in March for the main basin. The highest sea ice coverage characterised 2019/20, 2021/22 and 2014/15, which were also the seasons with the largest number of negative air temperature days in October – December.

The season 2019/20 was characterised by the lowest mean daily and monthly air temperatures. We observed a remarkable inter-annual variability in the sea ice coverage but at the nine-season scale we did not record any gradual trend of decreasing sea ice coverage. These high-resolution data can be used to e.g., better understand the spatio-temporal trends in the sea ice distribution in Hornsund, facilitate comparison between Svalbard fjords and improve modelling of nearshore wind wave transformation and coastal erosion.

**1 Introduction**

Sea ice plays a key role for the climate by controlling heat, moisture, gas and light transfer between the ocean and the atmosphere. It impacts wildlife and human activities in high latitudes (Maier and Stroeve, 2008). It modifies the wave energy transfer and, consequently, protects polar coasts from wave-driven erosion. Fast ice directly precludes waves from entering the foreshore and backshore (Rodzik and Zagórski, 2009), while drift ice attenuates wave energy, mostly at higher

frequencies (Barnhart et al., 2014; Nederhoff et al., 2022).



The Arctic annual mean sea ice extent decreased by 3.5-4.1% per decade between 1979 and 2012 (IPCC, 2019). At the same time the sea ice free season along the Arctic coastlines increased 1.5-3 times (Barnhart et al., 2014). In Fram Strait, west of Svalbard, increasing dominance of the Atlantic Water over the Polar Water elevated the sea surface temperatures contributing to an intensified melting of the sea ice transported in the East Greenland Current and delayed in situ ice growth (de Steur et al., 2023). In Svalbard itself despite the high inter-annual variability (Smith and Lydersen, 1991; Gerland and Renner, 2007) there is a general decline in sea ice extent and duration observed for various locations including Hornsund, Isfjorden, Kongsfjorden and Rijpfjorden (Muckenhuber et al., 2016; Pavlova et al., 2019; Johansson et al., 2020). Svalbard-wide fast ice decline rates of 100 km$^2$ yr$^{-1}$ was observed for 1973-2018 (Urbański and Litwicka, 2022). Fjords in western Svalbard experienced accelerated reduction in sea ice extent in 2000-2016 compared to 1980-2000 (Dahlke et al., 2020). Simulations suggest that in 2060-2069 the western fjords and northern Barents Sea will experience further reduction in sea ice concentration reflecting increasing sea surface temperatures (Hanssen-Bauer et al., 2020).

With decreasing extent and duration of the sea ice cover and increasing storminess, Arctic coasts are exposed to larger waves during extended time periods (Forbes, 2011; Overeem et al., 2011; Zagórski et al., 2015). This intensifies coastal flooding and erosion, posing infrastructure at a risk of damage (Casas-Prat and Wang, 2020). Therefore, it is critical to understand the seasonal patterns and longer-term (years to decades) trends in sea ice coverage at a local (fjord) scale, to better model wave energy delivery to the shores and, consequently, coastal hazards.

Satellite data have been used in sea ice mapping since 1979 (Parkinson and Cavalieri, 2008). Passive microwave-derived sea ice products have the longest temporal record but lack the high spatial resolution offered by synthetic aperture radar (SAR) imagery (Wang et al., 2016). Since May 2007, the Norwegian Meteorological Institute provides manually drawn ice charts for Monday-Friday for the Arctic Ocean including Svalbard (https://cryo.met.no/en/ice-service, access 24 July 2023). These maps make use of SAR images, most recently from the ESA Sentinel-1 missions, as well as in situ observations, and wind estimates. Several daily sea ice concentration products based on satellite images or models are available from the National Snow and Ice Data Center (https://nsidc.org/data, access 24 July 2023), Copernicus Marine Data Store (https://data.marine.copernicus.eu/products, access 24 July 2023), EUMETSAT Data Services (https://navigator.eumetsat.int/start, access 24 July 2023), Ifremer Wind and Wave Operation Center (Ardhuin, 2022) and the University of Bremen (Melsheimer and Spreen, 2019). There are variations between the products in the areas around Svalbard (Swirad et al., 2023a), which reflects their inconsistencies at the marginal ice zone i.e., at the vicinity of the sea ice edge (Andersen et al., 2007; Ozsoy-Cicek et al., 2017; Pang et al., 2018). This may be due to diverse sensitivity of algorithms to sea ice temperature, atmospheric effects and thin ice (Ivanova et al., 2015). In coastal areas the products may yield errors related to mixed land and water surface within the sensor footprint (Cavalieri et al., 1999). The product resolution of 1 to 25 km is often too low to analyse details in seasonal and inter-annual changes along the coast.



We use the method developed by Johansson et al. (2020) to make near-daily binary ice/open water maps of Hornsund fjord in southern Svalbard based on the full Sentinel-1A/B SAR archive over nine seasons (2014/15-2022/23). We describe spatial and temporal distribution in the sea ice coverage and combine the results with earlier (1999/2000-2013/14) study of Muckenhuber et al. (2016) to identify longer-term (24 years) trends.

**2 Study area**

Hornsund fjord is located in south-western Spitsbergen, Svalbard. Its opening to the Greenland Sea in the west is ~12 km wide and defined by Worcestepynten (north) and Palffyodden (south). The fjord is ~30 km long and can be divided into the main basin and secondary bays as indicated in Fig. 1b. The inner (eastern) part, Brepollen is ~13 km long and separated from the main basin of Hornsund by ~2 km wide opening between Treskelodden (north) and Meranpynten (south). Burgerbukta is

a ~8 km long bay in northern Hornsund delimited by Hyrneodden (east) and Brattpynten (west). It branches into Austre (eastern) and Vestre (western) Burgerbukta. In southern Hornsund, Samarinvågen is ~7 km long and defined by Meranpynten (east) and Traunpynten (west) (Fig.1; https://toposvalbard.npolar.no/, access 4 July 2023).

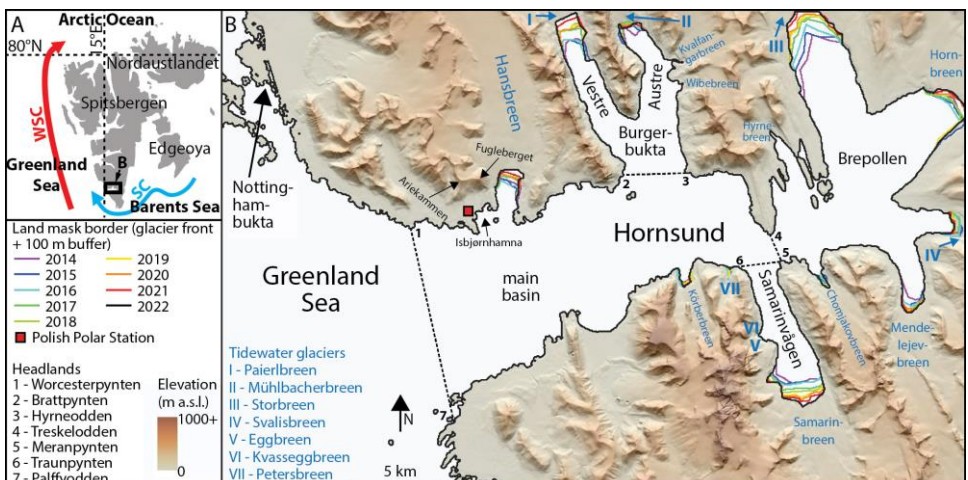

**Figure 1: Figure 1. Study area: a) Svalbard, b) Hornsund. WSC = West Spitsbergen Current; SC = Sørkapp Current. DEM courtesy of the Polar Geospatial Center.**

The average fjord depth is ~100 m but it reaches 200-250 m in the main basin. The circulation is anticlockwise with the inflow from the southwest and outflow to the northwest, and the average tidal range is 0.75 m (Herman et al., 2019). The influence of the West Spitsbergen Current that carries warm Atlantic water from south-southwest is limited by the strong

influence of the Sørkapp Current that brings cold water masses from the Barents Sea (Fig. 1a). Annual along-fjord CTD measurements conducted in July 2001-2015 showed surface layer temperatures of 3-6°C outside the fjord and in the main basin, and 0-3°C in Brepollen, where at depth >75-100 m the temperature was negative (Promińska et al., 2018).





Sea ice in Hornsund is either transported into the fjord from eastern Svalbard with the cold Sørkapp Current (blue line in Fig.
1a) or formed in situ as fast ice (Kruszewski, 2011). The ice drifting into the fjord from southwest typically first appears in
November – December, while the in situ fast ice onset is between November and March. The fjord is usually sea ice free by
the mid/end of June (Muckenhuber et al., 2016).

For the seasons 2005/06-2011/12 hand drawn maps of the sea ice extent, concentration and type (stage of development and
morphology) were made based on observations from slopes of Fugleberget and Ariekammen, and the Isbjørnhamna shore
(Fig. 1). These maps covered the main basin closest to the Polish Polar Station (PPS), showing seasonal and interannual
variability in the sea ice conditions, with the lowest sea ice coverage in 2005/06 and 2011/12 (Styszyńska and Kowalczyk,
2007; Styszyńska and Rozwadowska, 2008; Styszyńska, 2009; Kruszewski, 2010; 2011; 2012; 2013). Muckenhuber et al.
(2016) used satellite data to investigate the fjord-wide sea ice coverage over 15 seasons (1999/2000-2013/14). April was
characterised by the largest coverage of fast ice averaging 42% for all seasons and decreasing from an average of 53% in the
first six seasons to 35% in the following nine seasons. The 2011/12 and 2013/14 seasons had the lowest fast ice coverage
that at no point exceeded 40%.

The average daily mean air temperature at the PPS in northern Hornsund was -3.7°C for 1979-2018, with March being the
coldest (-10.2°C) and July the warmest (4.6°C) month. October to April had higher interannual air temperature variability
compared to the more consistent May to September. There is a trend of increasing mean annual air temperature by 1.14°C
per decade with the highest monthly increase of 2.1-2.4°C per decade in December, January and February (Wawrzyniak and
Osuch, 2020).

Wind conditions recorded at the PPS are strongly influenced by the shape of the Hornsund fjord, topography and coastal
location. In 1979-2018 the easterly winds dominated with the mean wind direction of 124° and the average wind speed of 5.5
m s$^{-1}$ with the minimum mean monthly of 4.0 m s$^{-1}$ in July and the maximum of 7.1 m s$^{-1}$ in February (Wawrzyniak and
Osuch, 2020). Long oceanic swell and mixed swell/wind sea from the south-southwest, and short locally generated wind
waves from the east determine wave conditions in the fjord. The mean significant wave height at the fjord opening is 1.2-1.3
m. It decreases to 0.25-0.43 m in the bays of the main basin, and further in the inner sections of the fjord. Waves are higher
and longer in the northern Hornsund and in the winter months (Herman et al., 2019; Swirad et al., 2023c). Preliminary
research suggested that sea ice present outside and inside the fjord attenuates wind wave energy, in particular at frequencies
>0.15 Hz (Swirad et al., 2023a).

Sixteen tidewater glaciers surround the fjord (Fig. 1b) with a total cliff length of 34.7 km in 2010. The average retreat rate of
glacier front was 70 ± 15 m yr$^{-1}$ in 2001-2010 compared to 45 ± 15 m yr$^{-1}$ for 128 glaciers in different parts of Svalbard.
Because of the glacier front retreat, the fjord area increased from ~188 km$^2$ in 1936 to ~303 km$^2$ in 2010 (Błaszczyk et al.,
2013). The average 2006-2015 freshwater input to the fjord was 2517 ± 82 Mt yr$^{-1}$ with contributions from glacier meltwater





runoff (39%), frontal ablation of tidewater glaciers (25%), precipitation over land excluding snow (21%), snowmelt (8%) and precipitation over fjord (7%) (Błaszczyk et al., 2019).

## 3 Methods

### 3.1 Dataset and pre-processing

Sentinel-1A/B SAR data between 14 October 2014 and 29 June 2023 were used in this study, including 2508 Extra Wide (EW) and 459 Interferometric Wide (IW) scenes that fully covered the Hornsund fjord. Automated segmentation and classification were performed on 2486 (84%) scenes that were of sufficient quality for processing and had both the HH and HV channels. If multiple scenes were available for the same date, only one was used with preference given to (i) the one that required no/less manual edition, (ii) IW scene, (iii) the EW scene with a mid-incidence angle range over Hornsund to ensure the best signal-to-noise ratio for both the HH and the HV channel. This resulted in the use of 2031 scenes, which constituted 68% of all scenes and 82% of the good quality scenes. The temporal frequency averaged 1.57 days with the maximum gap of 12 days (no scenes on 17-28 June 2016), and a generally higher frequency between April 2016 and December 2021 when both Sentinel-1A and Sentinel-1B were operating.

We processed the data in the same way as Johansson et al. (2020), where the scenes were geocoded on a fixed grid using 50 × 50 m pixel spacing with the grid extent of X: 499-544 km and Y: 8531-8559 km of WGS-84, UTM zone 33N using GSAR software (Larsen et al., 2006). For each scene three Geotiff rasters were created with the radar backscatter sigma naught values (HH and HV separate) and the incidence angle.

### 3.2 Image segmentation

We used the segmentation algorithm developed by Cristea et al. (2020) and adapted to the Svalbard fjord environment by Johansson et al. (2020). The algorithm accounts for the surface-specific intensity decay rate with increased incidence angle, meaning that the different segments contain areas with the same physical structure, as was further shown by Cristea et al. (2022). For each new segment the traditional Expectation Maximisation is performed, and Pearsons's style goodness-of-fit (GOF) test is carried out. If the test does not pass the GOF criterion, further splitting of the segments is carried out. Finally, a Markov random field-based contextual smoothing filtering is done to ensure simpler visual interpretation. For this study a maximum number of segments was set which was deemed sufficient to separate ice from open water. In 90% of the cases 5 segments were found to be suitable and for the remaining 10% a range from 3 to 7 segments was used. The challenging cases were typically those with topography-driven varying wind characteristics across the scene, and those where the SAR beam boundaries crossed the fjord (Johansson et al., 2020).

As a standard, both the HH and HV channels were used. The HV channel ensures good separation between open water and first year ice (Zakhvatkina et al., 2017). However, because of the low signal-to-noise ratio, the HV channel does not



necessarily improve the segmentation. At times only one of the channels was used to ensure the best possible ice/open water separation. During the segmentation the images were multi-looked by 3×3 and log-transformed (Johansson et al., 2020), and a land mask was used.

The land mask was created using the Norwegian Polar Institute's land shapefile (NPI, 2014). The mask was updated annually using the SAR scenes to account for glacier front retreats in the fjord. The scene from 1 July (or first available thereafter) was assumed to depict the furthest extent of the Hornsund glaciers after the front advance in spring (Błaszczyk et al., 2013). The 2014 mask was based on a RADARSAT-2 scene, and thereafter Sentinel-1 scenes were used. Each mask was applied to the following 1 July – 30 June period. Finally, a 100 m buffer was added along the entire coast with a larger buffer
area of 3.3 km$^2$ of shallow Nottinghambukta delta (Fig. 1b) to exclude the tidal zone.

### 3.3 Ice/open water classification and post-processing

The segmentation output was a Geotiff raster with ~5 discontinuous segments. The segments were manually ascribed as either 'ice' or 'open water' based on user experience and assumptions of (i) low backscatter values of open water with calm wind conditions and higher values for ice, and (ii) better separation of the two classes with HV channel which is less affected
by wind-roughed surfaces (Johansson et al., 2020). The method does not allow for the separation of drift, fast and glacier ice.

Of the 2031 scenes, 1432 (71%) did not require manual editing after class assignment to the segments. However, 333 (16%) scenes required minor manual change consisting of adding/removing polygons to/from the ice class, and 266 (13%) scenes required entirely manual ice polygon picking (without border drawing or polygon splitting) similar to that of Muckenhuber
et al. (2016). Manual editing was preceded by conversion of Geotiff rasters into polygons and followed by rasterizing. The classification resulted in near-daily binary maps of ice (1) and open water (0) available in the PANGAEA repository (Swirad et al., 2023b), together with a document containing details on the scenes and channels used, number of segments and the level of manual editing.

The sea ice extent (m$^2$) and coverage (%) were extracted for the entire Hornsund and separately for Burgerbukta, Brepollen, Samarnvågen and the main basin (as delimited in Fig. 1b) for the 2031 scenes (days). The time series is available as Table S1 in the Supplement. An annual onset and end of drift and fast ice was defined visually from the raw scenes and the binary maps. Analyses were performed on an annual basis starting in September (lowest sea ice coverage after Johansson et al., 2020) and ending in August. For instance, season 2014/15 spanned 1 September 2014 – 31 August 2015. The sea ice season,
however, started with the first arrival of the drift ice from the southwest on 29 December 2014 and ended on the last day with fast ice on 7 July 2015.



## 3.4 Validation

The method was validated in Johansson et al. (2020) by comparison between the classification results and the location of the fast ice edge tracked with a hand-held GPS, and by comparison to manually drawn maps from mountain Zeppelin above Ny-190 Ålesund. Since the same pre-processing, segmentation algorithm set-up and classification method were used, we consider these results also valid for our study in Hornsund. Additionally, we manually traced the fast ice edge in 20 Sentinel-2 optical images that were acquired on the same days as Sentinel-1 SAR scenes, had cloud cover below 10% and an unambiguous fast ice edge position without adjacent drift ice. We converted classified sea ice polygons into lines and extracted the fast ice edge, which we resampled at 5 m (0.1 px) spacing. In total, we compared 299.23 km fast ice edge (59,866 points) The mean 195 distance varied from 43 m ± 31 m to 131 m ± 177 m (average 80 m), equivalent to 0.85 ± 0.62 to 2.62 ± 3.53 px (average 1.59 px). Table S2 in the Supplement contains details on the validation.

## 3.5 Additional datasets

Muckenhuber et al. (2016) provided a time series of open water, fast ice and drift ice extent ($m^2$) in Hornsund for 1381 satellite images between 1 March 2000 and 8 June 2014. To enable direct comparison with our results, we calculated sea ice 200 coverage (%) as (fast ice extent + drift ice extent) / total extent $\times$ 100%.

Continuous meteorological data from the PPS are available at https://monitoring-hornsund.igf.edu.pl/ (access 5 July 2023). Mean daily air temperature is derived from 3-hourly automated measurements with Vaisala HMP45D / HMP155 probe at 2 m a.g.l. (~12 m a.s.l.) in the vicinity of the main buildings of the PPS, ca. 200 m from the shore. Sea water temperature 205 measurement is performed daily at 12:00 UTC at the shore next to the PPS boathouse using Elmetron thermometer PT-401. The reading is representative for the top 50 cm layer. Data gaps, mostly in winter months, are caused by presence of ice or strong winds.

## 4 Results

Binary ice/open water maps at 50 m resolution were made for the period 14 October 2014 to 29 June 2023 at the average 210 frequency of 1.57 days (Swirad et al., 2023b). The ice coverage varied in space and time during the inspected nine seasons (Fig. 2).





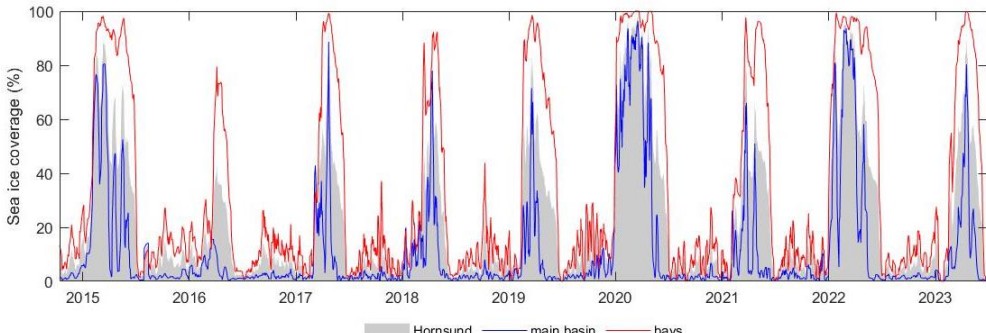

**Figure 2: Time series of sea ice coverage in Hornsund in 2014-2023 derived for the whole fjord (shadow), the main basin (blue line) and the bays (red line) smoothed over 5 scenes. Bays refer to Burgerbukta, Brepollen and Samarinvågen as delimited in Fig. 1.**

The first day of drifting sea ice entering the fjord from the southwest was between October and March. The fast ice onset was on average 24 days later, between December and March. The last day of drift ice was in May or June and occurred on average 20 days before the end of the fast ice season in June, except for 2014/15 (July) and 2015/16 (May). The length of the entire sea ice season (drift ice and fast ice) averaged 158 days, ranging from 105 (2015/16) to 246 days (2019/20). The length of the fast ice season was on average 134 days, ranging from 61 to 195 days also in 2015/16 and 2019/20, respectively 220 (Table 1).

**Table 1: Timing and duration of sea ice coverage in Hornsund in 2014-2023, and the mean coverage in the entire fjord and separately in the main basin and the bays (Burgerbukta, Brepollen and Samarinvågen as delimited in Fig. 1).**

| Season | First drift ice | Fast ice onset | Last drift ice | Fast ice end | Length of sea ice season (days) | Length of fast ice season (days) | Mean sea ice coverage (%) | | | | | |
| --- | --- | --- | --- | --- | --- | --- | --- | --- | --- | --- | --- | --- |
| | | | | | | | Entire sea ice season | | | Fast ice season | | |
| | | | | | | | Hornsund | main basin | bays | Hornsund | main basin | bays |
| 2014/15 | 29 Dec | 27 Jan | 13 Jun | 7 Jul | 191 | 162 | 49 | 27 | 74 | 54 | 30 | 81 |
| 2015/16 | 15 Feb | 30 Mar | 26 May | 29 May | 105 | 61 | 23 | 6.1 | 41 | 28 | 3.6 | 55 |
| 2016/17 | 1 Mar | 7 Mar | 18 May | 18 Jun | 110 | 104 | 41 | 18 | 66 | 43 | 18 | 70 |
| 2017/18 | 30 Dec | 29 Jan | 23 May | 4 Jun | 157 | 127 | 29 | 15 | 43 | 34 | 18 | 51 |
| 2018/19 | 25 Jan | 12 Feb | 3 Jun | 18 Jun | 145 | 127 | 40 | 14 | 67 | 45 | 16 | 76 |
| 2019/20 | 28 Oct | 18 Dec | 26 May | 29 Jun | 246 | 195 | 56 | 41 | 72 | 67 | 49 | 87 |
| 2020/21 | 2 Feb | 4 Feb | 8 Jun | 24 Jun | 143 | 141 | 39 | 14 | 64 | 39 | 14 | 65 |
| 2021/22 | 3 Dec | 28 Dec | 23 May | 22 Jun | 202 | 177 | 52 | 34 | 70 | 62 | 41 | 85 |
| 2022/23 | 11 Feb | 21 Feb | 24 May | 12 Jun | 122 | 112 | 45 | 19 | 71 | 48 | 21 | 77 |
| Average | 11 Jan | 4 Feb | 27 May | 17 Jun | 158 | 134 | 41 | 21 | 63 | 47 | 23 | 72 |

The average sea ice coverage in Hornsund during the entire sea ice season was 41% ranging from 23% in 2015/16 to 56% in 2019/20 (Table 1; Fig. A1). The coverage was lower in the main basin (average 27%, range 6.1%-41%) compared to the bays (average 63%, range 41%-74%), and of the three bays, Burgerbukta had the lowest ice coverage (average 50%)

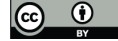



compared to Brepollen (67%) and Samarinvågen (69%) (Fig. 3). The sea ice coverage during the fast ice season was on average 6.5% higher than during the entire sea ice season and shows the same spatial and temporal trends (Table 1).

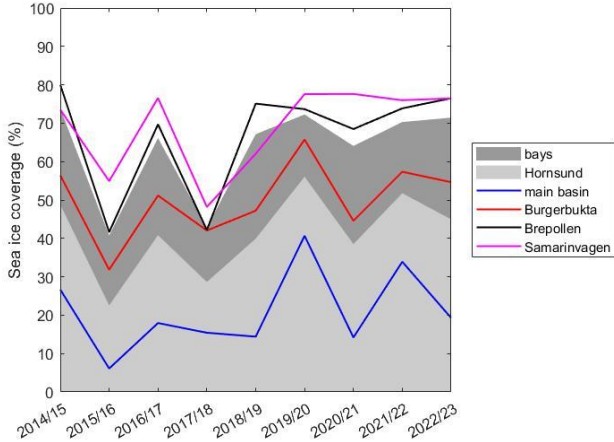


**Figure 3: Average sea ice coverage during the entire sea ice season (from the first day of drift ice to the last day of fast ice) for Hornsund and its parts in 2014-2023. Bays refer to Burgerbukta, Brepollen and Samarinvågen as delimited in Fig. 1.**

The highest sea ice coverage in Hornsund occurred in April with an average of 58%, followed by March (56%), May (40%) and February (36%). In the bays, the maximum coverage was in April (88%), May (75%), March (73%) and February

(47%). In the main basin, the highest average coverage characterised March (41%), April (31%) and February (26%) (Figs A2-A7). At no time during the entire study period did the mean monthly sea ice coverage in the main basin exceed that of the bays. Seasons 2014/15, 2019/20 and 2021/22 were characterised by an early (January – February) increase in sea ice coverage by 58-66% in the entire fjord and by 63-80% in the bays. In the 2019/20 the coverage remained relatively high for both the main basin and the bays, with the latter experiencing a mean monthly coverage of >90% over five consecutive

months (Fig. 4).

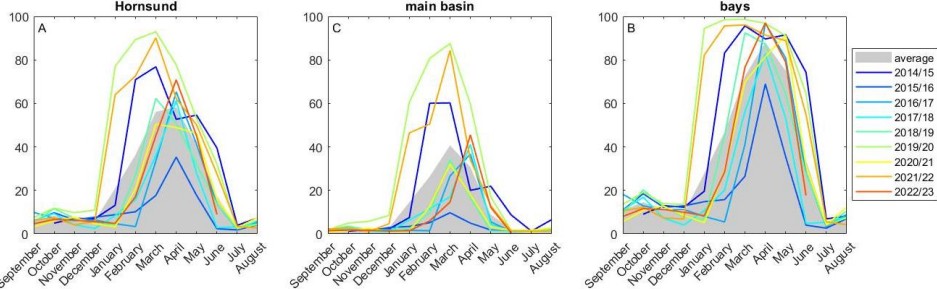

**Figure 4: Distribution of the mean monthly sea ice coverage (%) in Hornsund in 2014-2023 over: a) the entire fjord, b) the main basin, and c) the bays (Burgerbukta, Brepollen and Samarinvågen as delimited in Fig. 1).**

There was a secondary coverage peak around October that reached up to 20-40% for specific days (Fig. 2). The ice coverage

in summer and autumn was typically related to tidewater glacier calving, though occasionally its increase was caused by drift ice from the southwest such as on 7 August 2015.





Overall, an early onset of drift and fast ice, long sea ice season and high sea ice coverage of both the main basin and the bays characterised 2014/15, 2019/20 and 2021/22. The 2015/16 season was short with late drift and fast ice onset, and low sea ice

coverage. Other short seasons were 2016/17 and 2022/23, and that of low coverage was 2017/18 (Figs 2-4; Table 1).

Figures 5 and 6 show changes in the sea ice coverage in Hornsund for 24 seasons. It is a compilation of the present study and that of Muckenhuber et al. (2016). In the period of 1 March 2000 – 14 November 2005 only optical images were used which made ice detection during the dark season (November – February) impossible. Since 15 November 2005 Envisat ASAR (150

m resolution) scenes facilitated more regular temporal coverage throughout the year, though the resolution might have affected the retrieved ice coverage. From 1 November 2011 near-daily RADARSAT-2 (50 m resolution) data were used throughout the sea ice season (Fig. 5). In earlier years (1999/2000-2010/11) the sea ice season was generally longer, extending to the second half of June – beginning of July, and the onset was relatively early (November – December; identifiable only since the SAR imagery became available) (Fig. 5). Over the 24-year period, seasons 2005/06, 2013/14 and

2015/16 were characterised by the lowest sea ice coverage that never exceeded 80%. The 2011/12 season was short with inconsistent coverage at the short-term (days/weeks) scale. Seasons with high sea ice coverage were 2007/08-2010/11, 2012/13 (Muckenhuber et al., 2016), and 2014/15, 2019/20 and 2021/22 (Fig. 6).

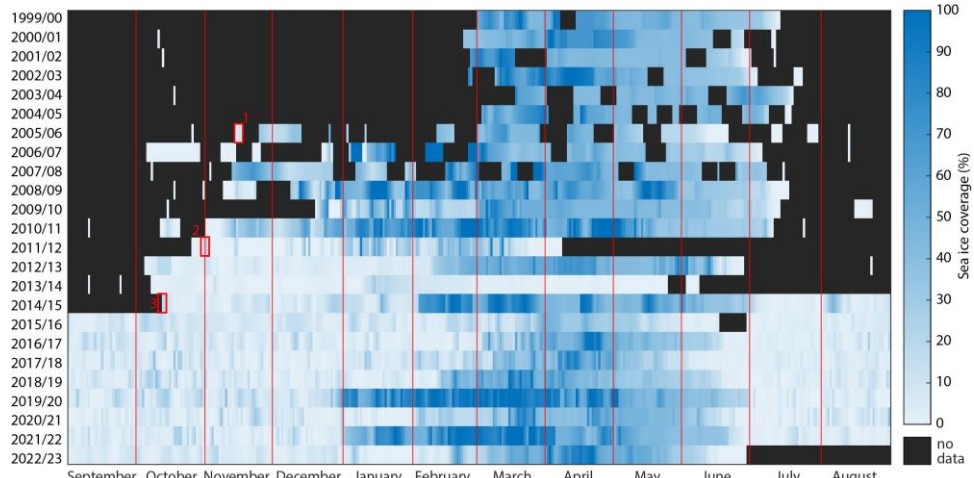

**Figure 5: Sea ice coverage in Hornsund in 2000-2023. Data from 1 March 2000 – 8 June 2014 were adapted from Muckenhuber et**
**al. (2016) and these from 14 October 2014 – 29 June 2023 were processed in this study. X-axis represents days of the season starting 1 September (as in Johansson et al., 2020) and Y-axis represents seasons. Linear interpolation over max 7 days was applied. First SAR scenes used: 1 – Envisat ASAR (15 November 2005), 2 – RADARSAT-2 (1 November 2011), 3 – Sentinel-1A/B (14 October 2014).**



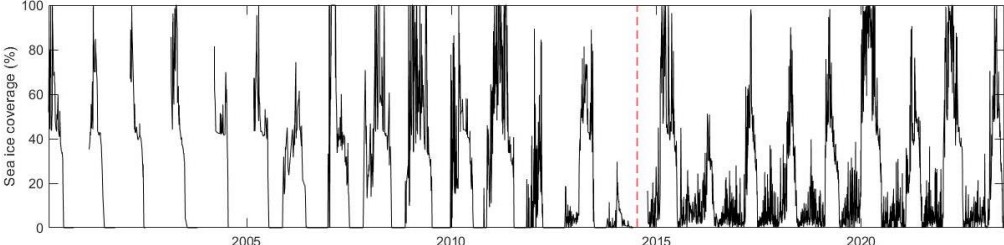

**Figure 6: Sea ice coverage in Hornsund in 2000-2023. Red dashed line separates the data adapted from Muckenhuber et al. (2016) and these from the present study. Note that different processing resulted in inclusion of glacier ice in our study, unlike in Muckenhuber et al. (2016), and that there are data gaps in the dark season (November – February) prior to the 2005/06 season caused by the use of optical imagery only.**

The lowest mean daily and monthly air temperature occurred in 2019/20 (Fig. 7; Table 2), which was the longest analysed sea ice season with the highest mean sea ice coverage (Table 1). The seasons with the below-average mean monthly air temperatures were 2022/23, 2014/15, 2017/18 and 2021/22, two of which (2014/15 and 2021/22) were also 'icy'. The three seasons with the highest sea ice coverage (2014/15, 2019/20 and 2021/22) overlapped with the years of the highest number of negative air temperature days in October – December but did not stand out in terms of later (January – June) air temperature or the number of positive air temperature days in the preceding year. The 2015/16 season with lowest sea ice duration and coverage had exceptionally high minimal mean monthly air temperature (-5.5°C), and similarly to the low sea ice coverage season 2017/18, it had an early onset of positive air temperatures in spring. The short sea ice season 2016/17 had the highest number of positive air temperatures in the preceding year and, consequently, the latest onset of negative air temperatures in autumn. The short sea ice season 2022/23 did not stand out in terms of air temperature variables included here. The longest negative sea water temperature period was in the high sea ice coverage season 2014/15, and the shortest – in the short sea ice season 2016/17, but there are no clear trends for other seasons (Table 2).

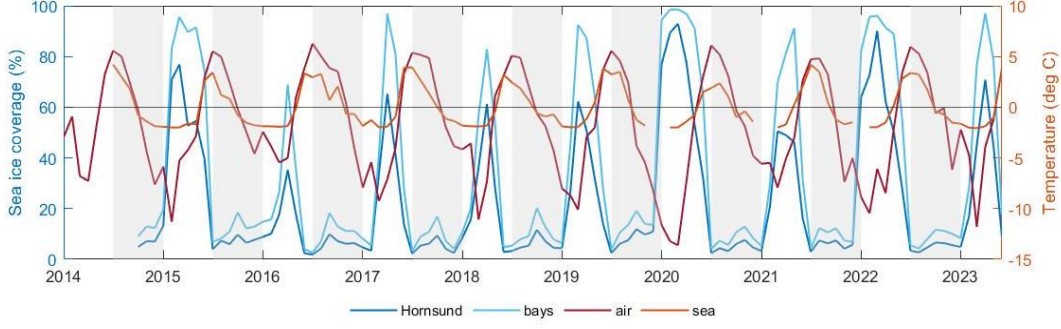

**Figure 7: Mean monthly sea ice coverage in Hornsund and the bays (Burgerbukta, Brepollen and Samarinvågen as delimited in Fig. 1) and air and sea water temperature in 2014-2023. The meteorological data were downloaded from https://monitoring-hornsund.igf.edu.pl/ (access 5 July 2023).**

290



**Table 2: Air and sea water temperature characteristics in Hornsund in 2014-2023. The meteorological data was downloaded from https://monitoring-hornsund.igf.edu.pl/ (access 5 July 2023).**

| Season | Min daily air temperature (ºC) | Min monthly air temperature (ºC) | Number of days with negative air temp Sep-Dec | Number of days with negative air temp Jan-Mar | Number of days with negative air temp Apr-Jun | Number of days of positive air temp days in the preceding year | Negative sea water temp timing | Negative sea water temp duration (length in days) |
|---|---|---|---|---|---|---|---|---|
| 2014/15 | -19.3 | -11.3 | 77 | 80 | 52 | 168 | 23 Sep – 3 Jun | 254 |
| 2015/16 | -16.1 | -5.5 | 61 | 75 | 37 | 170 | 24 Sep – 22 May | 242 |
| 2016/17 | -16.6 | -9.2 | 37 | 79 | 57 | 212 | 26 Oct – 24 May | 211 |
| 2017/18 | -15.0 | -11.1 | 58 | 79 | 35 | 170 | 6 Oct – 22 May | 229 |
| 2018/19 | -20.8 | -10.1 | 65 | 88 | 41 | 186 | 11 Sep – 14 May | 246 |
| 2019/20 | -24.4 | -13.6 | 87 | 90 | 41 | 147 | 10 Oct – 6 Jun | 241 |
| 2020/21 | -18.2 | -7.9 | 60 | 83 | 53 | 173 | 4 Oct – 23 May | 232 |
| 2021/22 | -20.1 | -10.5 | 78 | 83 | 50 | 149 | 30 Sep – 15 May | 228 |
| 2022/23 | -19.8 | -11.8 | 61 | 77 | 43 | 169 | 12 Oct – 30 May | 231 |
| Average | -19.4 | -10.1 | 65 | 82 | 45 | 172 | 3 Oct – 25 May | 235 |

## 5 Discussion

### 5.1 Consideration of the method

The segmentation algorithm of Cristea et al. (2020) facilitated efficient mapping of the sea ice in the Hornsund fjord. The method differed from that of Muckenhuber et al. (2016) as it usually (87%) did not require manual classification of single ice polygons but rather classifying small number (3-7) of discontinuous segments into ice or open water, which is likely quicker and more accurate.

We consistently used Sentinel-1A/B SAR data, where both the HH and HV channels were used in 1858 scenes, the HH only in 83 scenes and the HV only in 90 scenes. We observed that rough water conditions often resulted in ambiguous segmentation in the HH channel, whereas the HV channel provided improved segmentation, the problem persistent for the Envisat ASAR data with only HH data available. Thin and wet ice detection was sometimes hampered, particularly in the HV channel due to the lower power return. The HV channel also suffers from beam boundaries created by the TOPS-mode of Sentinel-1, which may make interpretation challenging.

The erroneous detection of ice may be related to radar shadowing and layover caused by topography, inclusion of shallow areas during low tide (thought this was addressed by masking 100 m at the coast with a larger mask in Nottinghambukta), atmospheric, wind and ocean wave effect, and temporal objects such as ships (Jones, 2023). On the other hand, the erroneous lack of detection may occur when sea ice is thin and wet, and difficult to separate from open water (Johansson et al, 2020). We addressed such situations during the manual editing of 29% of the scenes, though all errors could not be removed.



## 5.2 Types of ice

The semi-automated classification approach did not allow for separating drift, fast and glacier ice, as the backscatter
signatures are largely similar. We observed the lowest ice coverage in July followed by a gradual increase with a peak
around October (Fig. 7). We interpret it as the tidewater glacier calving activity, a finding consistent with Błaszczyk et al.
(2013) who recorded the most intense calving in August – November. However, a caution should be taken when using our
dataset for glacier ice analysis, as (i) brash ice, growlers and bergy bits may only be detected when grouped, (ii) during the
sea ice period the glacier ice is blended with the sea ice, and (iii) the 100 m coastal buffer of the land mask filters out glacier
ice accumulated in bays and glacier forefields.

We visually established the onset and end of the presence of drift and fast ice. The drift ice originates from two sources – it
is brought from the southwest by the Sørkapp Current or broken off from the fast ice formed in situ. Drifting sea ice is
therefore more common in the main basin of Hornsund than in the bays. The differences in sea ice timing and coverage in
the main basin and the bays (Fig. 4) agree with the observations of drift and fast ice, respectively, by Muckenhuber et al.
(2016), e.g., on the earlier onset of the drift ice and the peak timing of the two types. The bays create the best locations for
the fast ice formation because of the shelter from waves and wind. Of the three considered bays, Burgerbukta had the lowest
(and most changing at the seasonal scale) sea ice coverage which might be related to the higher wind wave energy in
northern Hornsund directly related to the long oceanic swell direction (Herman et al., 2019; Swirad et al., 2023c). In
Brepollen, the narrow opening and the presence of a sill restricts water exchange with the main basin, and relatively cold and
low salinity water persists throughout the year (Promińska et al., 2018). Good conditions for fast ice formation and
persistence in Samarinvågen may be related to its calm water (south shore of Hornsund), low incoming wind wave energy,
and bay's narrow and elongated shape.

## 5.3 Seasonal trends

An insight into seasonal patterns of the sea ice conditions was provided by visual observations and ice situation sketching
during seven consecutive seasons (2005/06-2011/12) at the vicinity of the PPS (Styszyńska and Kowalczyk, 2007;
Styszyńska and Rozwadowska, 2008; Styszyńska, 2009; Kruszewski, 2010; 2011; 2012; 2013). The observations suggested
that the main basin experiences higher variability of sea ice extent and concentration compared to Burgerbukta, Brepollen
and Samarinvågen bays (visible also in Fig. 2), which is related to the direct impact of instantaneous wind conditions and ice
situation outside Hornsund on the situation in the main basin (Kruszewski, 2012). The pack ice drifting from the southwest
in Sørkapp Current ranges from grey ice (10-15 cm thick) to thick first year ice (<2 m) (Styszyńska and Kowalczyk, 2007;
Styszyńska, 2009). Presence of the drift ice in Hornsund is dynamic at a daily to weekly scale, but in 2007/08 it remained in
the fjord for nine consecutive weeks (Styszyńska, 2009). Dominating strong easterly winds can make drift ice concentration
in the main basin of Hornsund lower than that of the open Greenland Sea (Kruszewski, 2011). The short-term variability in
sea ice coverage in Hornsund is visible for the 24 seasons in Fig. 6.





Fast ice forms in the major bays, and occasionally along less indented sections of the coasts. It is often broken by strong winds and waves, and drifts towards the main basin. Because of temperature variability, particularly in autumn and early winter, ice formation may start two or three times, and is interchanged with ice melting and/or breaking at periods of higher
air temperatures and/or stronger winds. At its maximum in 2010/11 the fast ice covered most of the main basin, while in 2011/12 it only extended across a portion of Brepollen (Kruszewski, 2012; 2013). The short-term fluctuations in sea ice coverage are clearly visible in Fig. 2.

Glacier ice is present in the fjord year-round in the form of brash ice, growlers, bergy bits and occasionally icebergs
(Kruszewski, 2010). It decreases the temperature and salinity of the surface water along the shore of Isbjørnhamna. In autumn, between glacier ice, new sea ice forms in situ – frazil ice, grease ice, shuga, nilas and grey ice, and later grey-white and pancake ice. Along the coast of the main basin an ice foot forms as a combination of glacier and sea ice, and freezing of swash, splashes and snow. It remains at the sea-land border until June – July, protecting the beach from erosion and redistribution of the material by waves (Rodzik and Zagórski, 2009; Zagórski et al., 2015).

**5.4 Inter-annual trends**

Presence of drift ice in Hornsund depends on winds, currents and tides (Kruszewski, 2010). On the contrary, the in situ sea ice formation depends on the intensity of heat release to the atmosphere which increases with lower temperatures and higher wind speed (Styszyńska and Kowalczyk, 2007) that control the thickness of the sea water surface mixed layer (Promińska et al., 2018). However, at the nine-year scale, we do not observe an expected decrease in fast ice coverage (Figs 2 and 4) as a
reflection of the air warming trend observed at the PPS (Warzyniak and Osuch, 2020). The reason for this may be that the number of seasons studied here is too short for such long-term trends. Similarly, for a 14-season (1974/75-1987/88) Smith and Lydersen (1991) observed a considerable interannual variability in Hornsund fast ice duration ranging from 0 to 166 days prior to April 1st but not a gradual trend.

Notably, Muckenhuber et al. (2016) stated that sea ice declined during the 15-season time series with two seasons of low sea ice coverage (2011/12 and 2013/14) occurring at the end of the analysed period. In our study, however, in the second half of the monitoring period, two seasons (2019/20 and 2021/22) were characterised by the highest mean sea ice coverage (Table 1), while 2011/12 and 2013/14 remain the seasons of the lowest sea ice coverage in Hornsund in the 21st century (Fig. 5). 2012 was also the year of the lowest pan-Arctic sea ice extent (https://nsidc.org/arcticseaicenews/2023/09/arctic-sea-ice-
minimum-at-sixth/, access 26 September 2023). The annual fast ice extent analysis across Svalbard by Urbański and Litwicka (2022) showed that there is a longer-term (1973-2018) trend of decreasing total fast ice extent by an average of 100 km$^2$ yr$^{-1}$ which resulted in halving over this ~45-year period. The modelling suggested that air temperature (freezing degree-days) has 60% influence on the interannual extent and duration of the fast ice (Urbański and Litwicka, 2022). For the Fram Strait, de Steur et al. (2023) found a direct relationship between sea water temperature and sea ice concentration, but the
relationship is still to be determined for the Svalbard fjords.



There is, however, an overlap of higher air temperatures and lower sea ice coverage, and lower air temperatures and higher sea ice coverage. Warm seasons with little sea ice were 2005/06, 2011/12, 2013/14, 2015/16 and 2016/17. Cold seasons with a lot of sea ice were 2002/03, 2003/04, 2007/08, 2008/09, 2009/10, 2010/11, 2012/13, 2014/15, 2019/20 and 2021/22 (Figs 385    5-7; Styszyńska, 2009; Kruszewski, 2012; 2013; Muckenhuber et al., 2016; Promińska et al., 2018). This pattern reflects the Svalbard-wide fast ice extent (Urbański and Litwicka, 2022).

For 14 field monitoring seasons in Kongsfjorden (2002/03-2015/16) Pavlova et al. (2019) observed an abrupt change between high sea ice coverage in the first three seasons and reduced sea ice extent and duration in the following 11 seasons. 390    The minimum was reached in 2011/12, while in 2009/10 and 2010/11 the sea ice coverage was high (Pavlova et al., 2019). Similar abrupt change starting in 2005/06 was observed by Muckenhuber et al. (2016) for Isfjorden with the lowest sea ice coverage in 2011/12 and 2013/14. Based on SAR images, Johansson et al. (2020) documented low sea ice coverage in Kongsfjorden in consecutive seasons 2005/06-2007/08 and 2011/12-2018/19, and in Rijpfjorden in 2011/12, 2015/16 and 2017/18. These findings generally agree with the situation in Hornsund (Fig. 6). We suggest that the differences may be 395    related to the local conditions, such as currents, winds and fjord shape. Due to the interference of the Sørkapp Current with the West Spitsbergen Current, the hydrographic conditions in Hornsund may differ considerably from those of more northern fjords (Promińska et al, 2018).

## 5.5 Further considerations

We did not find relationships between sea water temperature at the foreshore of Isbjørnhamna and the sea ice coverage 400    (Table 2). Because of the in-person character of the water temperature measurement, the data are missing for 34% days in the period of 1 January 2014 – 30 June 2023. Days when no measurement was taken were characterised by the presence of ice and/or strong winds, and a note of the conditions was usually made in the report. When the fast ice was present, the water surface temperature could be assumed to be below the saltwater freezing point of -1.78℃, while when the sea/glacier ice cover was discontinuous it was probably close to the freezing point. Therefore, we generally treated no data as negative 405    temperature (see negative temperature timing and length in Table 2 despite the lack of data for a considerable part of the winter). Kruszewski (2011) observed that the foreshore sea water temperature was higher when the waves were smaller and during the low tide. Continuous mooring (see e.g., De Rovere et al., 2022 for Kongsfjorden) would allow measurements from a depth of ~20-50 m that would be relevant to the sea ice but independent from instantaneous weather/surface conditions. Such an approach was used by de Steur et al. (2023) who found a correlation between upper layer (55 m) water 410    temperature and sea ice concentration in western Fram Strait. For this study, we only extracted the timing and length of negative surface water temperatures, parameters likely independent from inaccuracies related to the measurement strategy and circumstances.



The interactions between sea ice and glaciers are multifaceted and still not fully explored. We have discussed the
contribution of glacier ice to the fjord ice, and its impact on sea surface temperature and local in situ sea ice formation.
However, presence of sea ice can also impact glacier dynamics by decreasing glacier front calving. In Hornsund, Pętlicki et
al. (2015) observed that pack ice restricted wave action at the front of the tidewater glacier Hansbreen in 2011, limiting notch
development around the waterline and, consequently, failure of the overhanging ice.

Our method does not allow a determination of drift ice concentration at the 50 m resolution, an important parameter in
modelling wind wave transformation. For instance, the Simulating WAves Nearshore (SWAN) model takes sea ice
concentration as an input to account for dissipation of wave energy caused by the presence of ice (Rodgers, 2019). High-
concentration drift ice on the Greenland Sea attenuates energy of long oceanic swell and mixed swell/wind sea, while in situ
ice precludes wind wave generation over the eastern parts of the fjord (Styszyńska and Rozwadowska, 2008). Ignoring the
sea ice in nearshore wave modelling may cause an overestimation of significant wave height, an underestimation of wave
period, and an overall overestimation of the total wave energy (Herman et al., 2019; Swirad et al., 2023a). Norwegian
Meteorological Institute ice charts and various sea ice concentration products provide this information; however, their
resolution is lower than that of nearshore wave models. We suggest down-scaling our binary maps to the resolution of a
wave model which would allow a calculation of the sea ice concentration at a lower-resolution scale under the assumption of
100% concentration in the 'ice' pixels and 0% concentration in the 'open water' pixels. In the future, splitting the sea ice
mapped here to smaller more homogeneous segments with field calibration and a classification method similar to Lohse et
al. (2019; 2020) could be tested.

The ice situation in Hornsund impacts transportation and landing, particularly troublesome during the PPS crew exchange in
June – July when cargo is loaded and unloaded in Isbjørnhamna. In July 2008, a 52 km wide belt of drift ice of varied
concentration made it difficult for the cargo ship to reach the PPS area (Styszyńska, 2009). In 2004 no hydrographic
measurements could be made during the annual R/V Oceania cruise in July because of the dense sea ice cover in the fjord
(Promińska et al., 2018). On the other hand, the ice protects the beach from erosion, and consequently the PPS boathouse
from damage. A later onset of the sea ice season may allow early winter storms to enter coastal zones, increasing the hazard
of flooding and erosion (Zagórski et al., 2015). High spatial and temporal resolution studies, such as this one, may help
better predict the probability of sea ice conditions in specific times of the year.

**6 Conclusions**

Sentinel-1A/B SAR imagery between 16 October 2014 and 29 June 2023 was used in combination with a segmentation
algorithm and a classification to create near-daily binary ice/open water maps at 50 m resolution for Hornsund fjord, south-
western Svalbard (Swirad et al., 2023b). The maps were used to identify spatial and temporal patterns in the sea ice
coverage, with the intention to be used as sea ice input into nearshore wind wave transformation and coastal erosion models.

Sea ice in Hornsund has a dual origin; it arrives from the southwest in the cold Sørkapp Current or is formed in situ. Over the analysed nine seasons, 2014/15 to 2022/23, the average length of the sea ice season was 158 days ranging from 105 to 246
days. Drift ice first appeared between October and March, preceding the onset of the fast ice by ~24 days. The end of the sea ice season was related to the removal of the fast ice from the inner parts of the fjord in June (July in 2014/15 and May in 2015/16), ~20 days after the last drift ice presence in the fjord.

The average total sea ice coverage over the sea ice season was 41%, ranging between 23 and 56%. It was lower in the main
basin (27%) and higher in the inner bays: Burgerbukta, Brepollen and Samarinvågen (63%). Of the bays, Burgerbukta had the lowest (50%) and Samarinvågen the highest (69%) average sea ice coverage. We suggest that this is related to the differences in wave energy delivered to the northern vs southern shores of Hornsund with long oceanic swell and mixed swell/wind sea from the south-southwest affecting the northern shores which contributes to sea ice breaking at the relatively wide opening of Burgerbukta.

The highest sea ice coverage characterised March for the main basin and April for the bays. Because of the arrival of the drift ice from the southwest and a preferential persistence of the fast ice in the calmer-water bays, we suggest that this timing reflects the peak occurrence of drift ice (March) and fast ice (April). There is a secondary peak around October which we interpret as glacier ice from calving of tidewater glaciers.

Our data show high interannual variability. The highest sea ice coverage characterised 2019/20, 2021/22 and 2014/15, seasons that also had the largest number of negative air temperature days in October – December. The 'iciest' 2019/20 was characterised by the lowest mean daily and monthly air temperatures. At the nine-season scale we were not able to identify any gradual trend of decreasing sea ice coverage.

Combining our results with the previous study of Muckenhuber et al. (2016) gives a full picture on sea ice coverage in Hornsund in the first 24 years of the 21$^{st}$ century. The lowest sea ice coverage characterised seasons 2011/12 and 2013/14. Generally, shorter sea ice seasons and lower sea ice coverage coincided with higher air temperatures, while long seasons and high sea ice coverage occurred in cold years.

The results further the understanding of the sea ice extent, duration and timing in a High Arctic fjord environment, and facilitate between-site comparison on changing sea ice conditions in Svalbard. The binary ice/open water maps can be further used to improve nearshore wind wave transformation modelling and risk assessment for the coastal zone.





**Appendix: Extended figures**

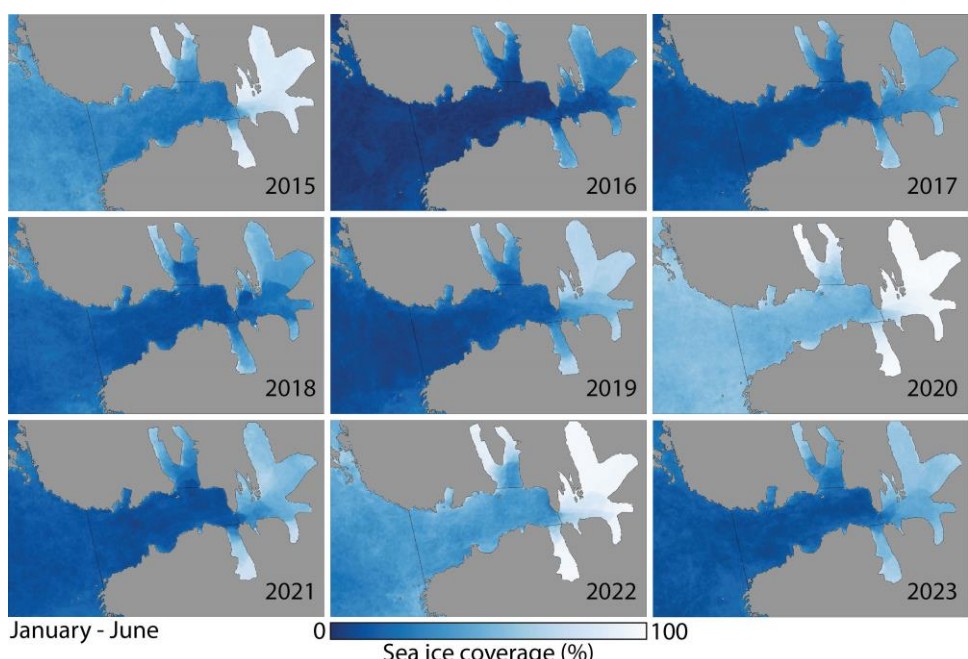


**Figure A1: Mean sea ice coverage in Hornsund in January – June 2015-2023.**

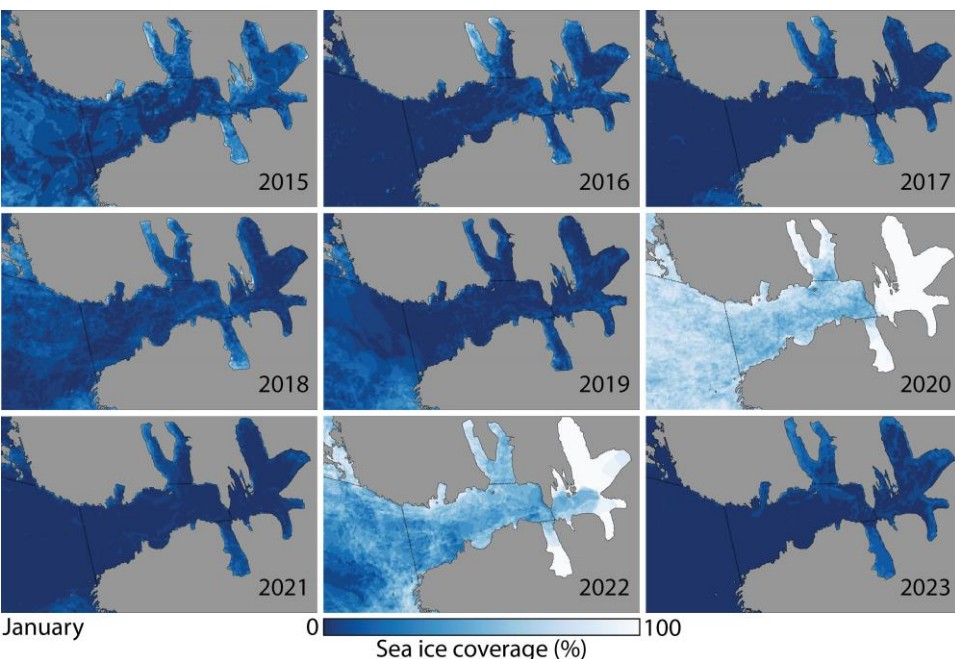

**Figure A2: Mean sea ice coverage in Hornsund in January 2015-2023.**





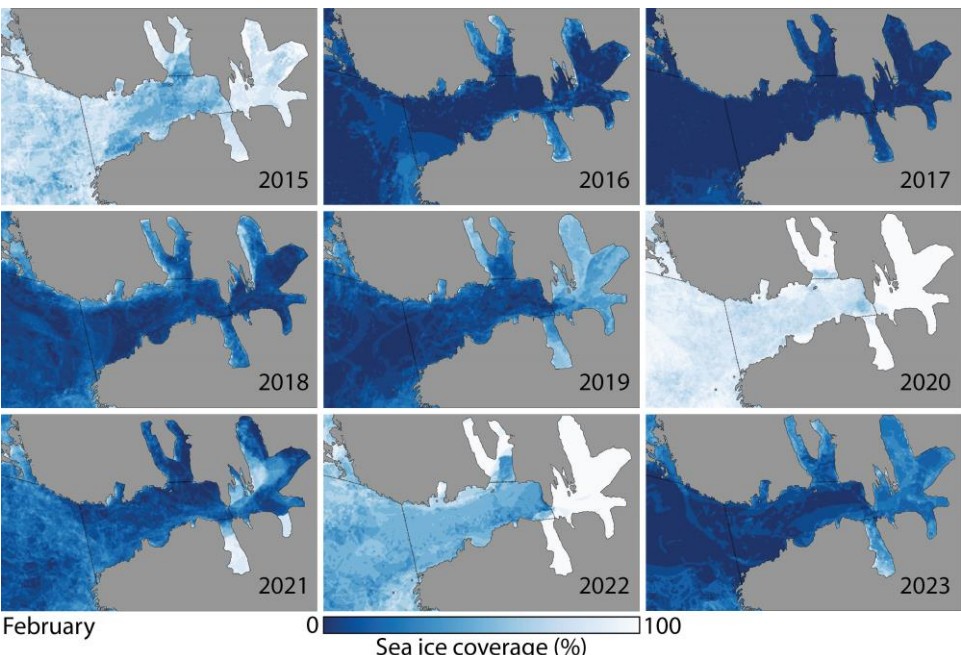

**Figure A3: Mean sea ice coverage in Hornsund in February 2015-2023.**

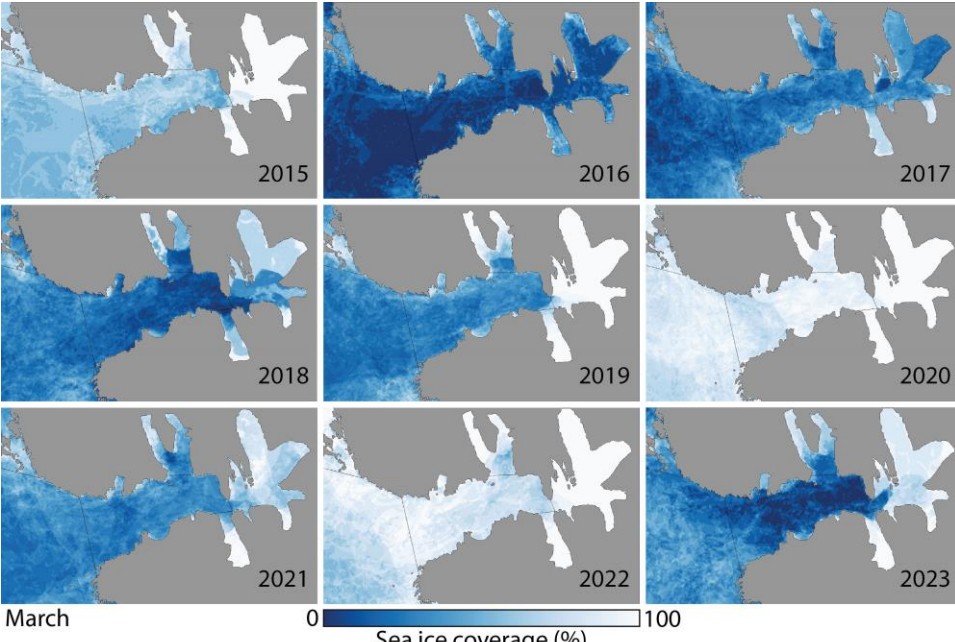

**Figure A4: Mean sea ice coverage in Hornsund in March 2015-2023.**



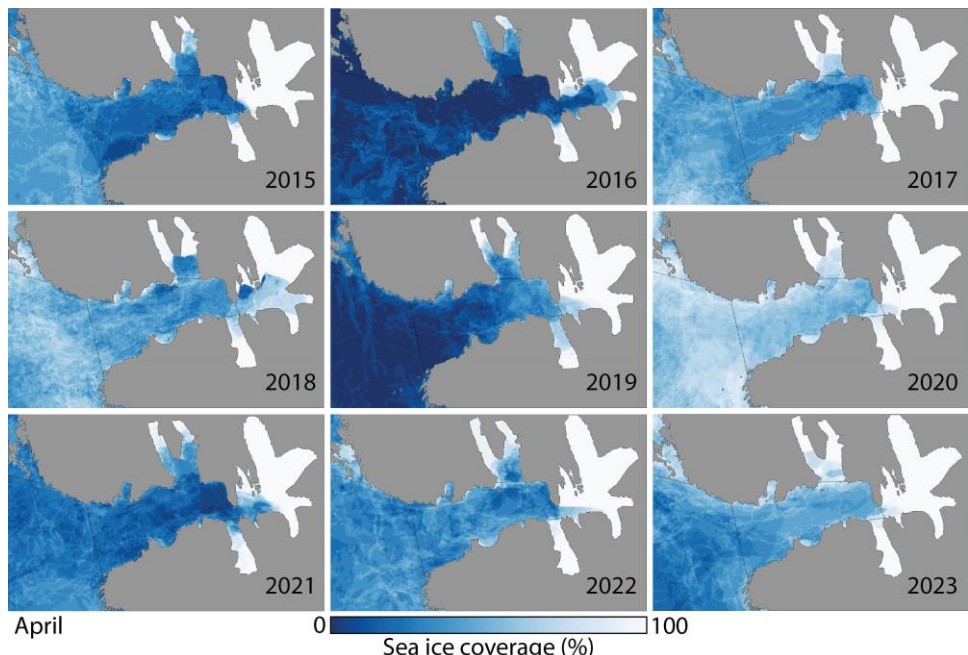

**Figure A5: Mean sea ice coverage in Hornsund in April 2015-2023.**

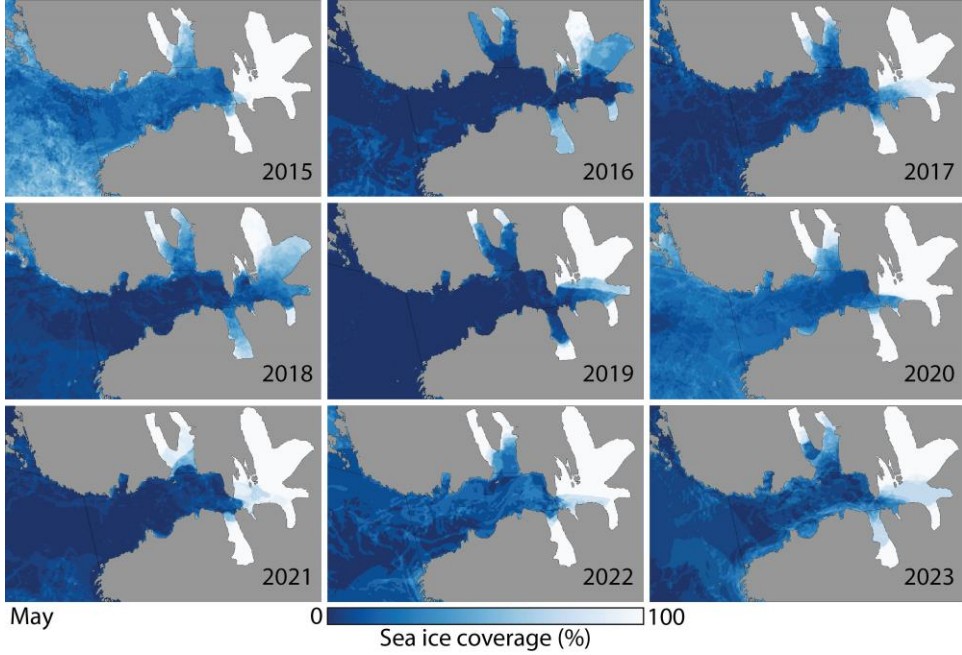


**Figure A6: Mean sea ice coverage in Hornsund in May 2015-2023.**

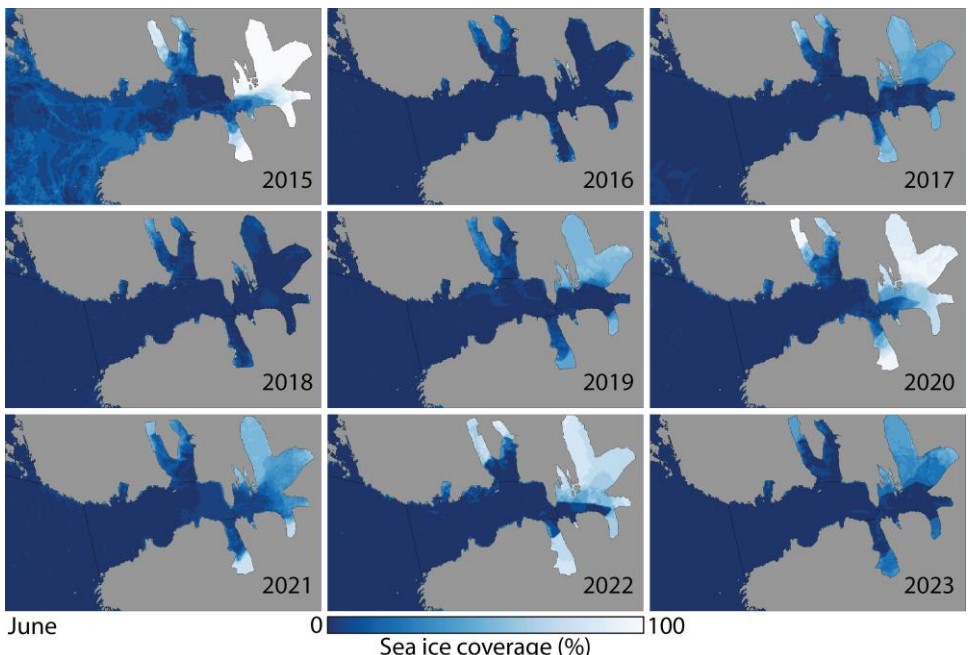

**Figure A7: Mean sea ice coverage in Hornsund in June 2015-2023.**

## Data availability

Binary ice/open water maps are available in the PANGAEA repository (https://doi.pangaea.de/10.1594/PANGAEA.963167; Swirad et al., 2023b). Table S1 in the Supplement is the summary of sea ice extent ($m^2$) and coverage (%) for Hornsund and its parts (as delimited in Fig. 1b) based on the binary maps.

## Author contributions

ZMS conceptualised the study, secured the funding and compiled the environmental data. EM pre-processed the SAR
scenes. AMJ developed the image processing method. ZMS processed and analysed the data with the help of AMJ. All authors interpreted the results and wrote the manuscript.

## Competing interests

None of the authors has any competing interests.

## Acknowledgements

We thank Anthony Doulgeris for discussion on the segmentation algorithm and the Polar Polish Station crew for maintaining the meteorological monitoring. We acknowledge the European Union's Earth observation program Copernicus which freely provides the Sentinel-1 and Sentinel-2 data (https://scihub.copernicus.eu).



**Financial support**

This study was funded by the National Science Centre of Poland (grant no. 2021/40/C/ST10/00146). AMJ was supported by
CIRFA and the Research Council of Norway under Grant 237906. EM was partly supported by the Research Council of
Norway under the project Svalbard Integrated Arctic Earth Observing System – Infrastructure development of the
Norwegian node (SIOS-InfraNor Project No. 269927).

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
