# Peer review of "Extent, duration and timing of the sea ice cover in Hornsund, Svalbard in 2014-2023"

_EGUsphere, 2023_

## Referee Comment (RC1)

**Extent, duration and timing of the sea ice cover in Hornsund, Svalbard in 2014-2023**

Zuzanna M. Swirad, A. Malin Johansson, Eirik Malnes

**General comments**

In this paper sea ice coverage in Hornsund fjord, Svalbard is investigated. Sea ice coverage was estimated by classifying Sentinel-1 SAR imagery to binary open water – sea ice maps at 50 m resolution. SAR imagery classification includes image segmentation following a previous study by Johansson et al. (2020) (i.e., not developed in this study) and manual classification of image segments to open water and sea ice. The temporal coverage of S-1 SAR imagery was nine ice seasons from 2014/15 to 2022/23. Validation of the open water – sea ice map was conducted by manually tracing fast ice edge in 20 Sentinel-2 images, and good co-incidence of S-1 vs. S-2 fast ice edge was found. Error sources for the classification, e.g. thin and wet sea ice mixed with open water, were also discussed. Supporting data include in-situ air and sea water temperature from the Polish Polar Station (PPS).

The maps were used for statistical analyses of spatial and temporal patterns in the sea ice coverage in the entire Hornsund fjord and its main basin and three major bays; including interannual variability and annual onset and end of drift and fast ice. For example, a large inter-annual variability in the sea ice coverage was observed, and at the nine-season scale there was no gradual trend of decreasing sea ice coverage.

The S-1 sea ice maps were also combined with maps from Muckenhuber et al. (2016) resulting sea ice coverage maps for 24 seasons; from 1999/00 to 2022/23. Seasons with low and high sea ice coverage were identified, again with no coverage trend was present, but there is an overlap of higher air temperatures and lower sea ice coverage, and lower air temperatures and higher sea ice coverage.

As the authors say this paper increases the understanding of the sea ice extent, duration and timing in a High Arctic fjord environment, and facilitate between-site comparison on changing sea ice conditions in Svalbard.

In general, the study set up with data acquisitions and data processing is well presented and conducted, and results which are based on large amount data are nicely presented and discussed. Conclusions of the study are well based on the statistical analyses and observations in the dataset. The paper is also very well written. I did not find any major shortcomings in the paper, and below are only few specific comments.

**Specific comments**

l. 26: "Sea ice plays a key role for the climate by controlling heat, moisture, gas and light transfer between the ocean and the atmosphere. It impacts wildlife and human activities in high latitudes (Maier and Stroeve, 2008)."

l. 50: "Passive microwave-derived sea ice products have the longest temporal record but lack the high spatial resolution offered by synthetic aperture radar (SAR) imagery (Wang et al., 2016)."

These are not the original references here, and references can be removed.

l. 45: Reference Overeem et al. 2011 is missing from the reference list.

Figures 1, 2, 4, 5 and 6 should have larger size, now difficult to see details.

l. 477: "The binary ice/open water maps can be further used to improve nearshore wind wave transformation modelling and risk assessment for the coastal zone."

This usage of the maps should be also described in Introduction Section.

You could show an example of original, segmented and classified SAR images. Preferably together with a Sentinel-2 image.

---

## Author Response (AR1)

**Reviewer 1**

**General comments**

**In this paper sea ice coverage in Hornsund fjord, Svalbard is investigated. Sea ice coverage was estimated by classifying Sentinel-1 SAR imagery to binary open water – sea ice maps at 50 m resolution. SAR imagery classification includes image segmentation following a previous study by Johansson et al. (2020) (i.e., not developed in this study) and manual classification of image segments to open water and sea ice. The temporal coverage of S-1 SAR imagery was nine ice seasons from 2014/15 to 2022/23. Validation of the open water – sea ice map was conducted by manually tracing fast ice edge in 20 Sentinel-2 images, and good co-incidence of S-1 vs. S-2 fast ice edge was found. Error sources for the classification, e.g. thin and wet sea ice mixed with open water, were also discussed. Supporting data include in-situ air and sea water temperature from the Polish Polar Station (PPS). The maps were used for statistical analyses of spatial and temporal patterns in the sea ice coverage in the entire Hornsund fjord and its main basin and three major bays; including interannual variability and annual onset and end of drift and fast ice. For example, a large inter-annual variability in the sea ice coverage was observed, and at the nine-season scale there was no gradual trend of decreasing sea ice coverage. The S-1 sea ice maps were also combined with maps from Muckenhuber et al. (2016) resulting sea ice coverage maps for 24 seasons; from 1999/00 to 2022/23. Seasons with low and high sea ice coverage were identified, again with no coverage trend was present, but there is an overlap of higher air temperatures and lower sea ice coverage, and lower air temperatures and higher sea ice coverage. As the authors say this paper increases the understanding of the sea ice extent, duration and timing in a High Arctic fjord environment, and facilitates between-site comparison on changing sea ice conditions in Svalbard.**

**In general, the study set up with data acquisitions and data processing is well presented and conducted, and results which are based on large amount data are nicely presented and discussed. Conclusions of the study are well based on the statistical analyses and observations in the dataset. The paper is also very well written.**

Thank you.

**I did not find any major shortcomings in the paper, and below are only few specific comments.**

**Specific comments**

**l. 26: "Sea ice plays a key role for the climate by controlling heat, moisture, gas and light transfer between the ocean and the atmosphere. It impacts wildlife and human activities in high latitudes (Maier and Stroeve, 2008)." l. 50: "Passive microwave-derived sea ice products have the longest temporal record but lack the high spatial resolution offered by synthetic aperture radar (SAR) imagery (Wang et al., 2016)." These are not the original references here, and references can be removed.**

We have removed both references.

**l. 45: Reference Overeem et al. 2011 is missing from the reference list.**

Thank you for spotting it. We have added it to the reference list.

**Figures 1, 2, 4, 5 and 6 should have larger size, now difficult to see details.**

We have increased the quality of the figures by exporting them in higher resolution. We increased their sizes in the manuscript.

**l. 477: "The binary ice/open water maps can be further used to improve nearshore wind wave transformation modelling and risk assessment for the coastal zone." This usage of the maps should be also described in Introduction Section.**

We have modified the third paragraph of the introduction to include this information. It now reads:

*"[...] Therefore, it is critical to understand the seasonal patterns and longer-term (years to decades) trends in sea ice coverage at a local (fjord) scale, to better model nearshore wind wave transformation and wave energy delivery to the shores and, consequently, more accurately assess coastal hazards and risks."*

**You could show an example of original, segmented and classified SAR images. Preferably together with a Sentinel-2 image.**

Good idea. We have added such a figure (now Fig. 2).
* * *
**Reviewer 2**

**General comments**

**In this paper, high resolution SAR imagery was used to create binary sea ice/open water maps to estimate the annual and interannual sea ice coverage of Hornsund fjord, Svalbard. Combined with maps from Muckenhuber, the study period was expanded to 24 seasons (2000-2023). The results showed that there was significant interannual variability of sea ice coverage in the study area and no coverage trend was present. Serval seasons with high and low sea ice coverage were identified, corresponding to lower air temperatures and high sea ice coverage, respectively.**

**In general, this study was well designed, and the paper is reasonably organized. The method, datasets and conclusions could be helpful to better understand the sea ice temporal and spatial characters in small scale. There are no major shortcomings in this paper.**

Thank you.

**Here are few specific comments.**

**Specific comments**

**Figure 1 B): The geographical scope of study area could be larger, so that readers can better understand the general situation of study area. The number of headlands can use larger font size or in a colored font.**

We have left the extent of panel b unchanged because it is exactly the extent of the binary maps which makes interpretation of binary maps easier, particularly in terms of increasing study area as glaciers retreat. However, we increased the quality and the size of the entire figure which makes reading panel a (context location) easier. As suggested, we have increased the size of the headland numbers.

**Line 194 ", which we resampled at 5 m (0.1 px) spacing." Why "5 m (0.1 px) spacing" was used?**

We have clarified the text which now reads:

*"To measure distance between lines of uneven nod distributions, we resampled both lines at 5 m (0.1 px) spacing and applied 'Distance to Nearest Hub (Points)' tool in QGIS."*

**Line 295. The method and data used in this study are different from that of Muckenhuber. How to consider the impact of using two datasets on the sea ice cover analysis in 2000-2023.**

Thank you for this comment. We have added a paragraph in section 5.4 that directly lists the key differences of the studies and their potential impact on analysis:

*"To expand the analysis and identify longer-term trends, we combined our study with that of Muckenhuber et al. (2016) which covered 15 preceding sea ice seasons. Three differences in datasets and/or methods need to be considered when interpreting the merged time series. Firstly, prior to season 2005/06 only optical images were used so it is impossible to analyse the sea ice season start and length, and winter-time coverage and its variability. Secondly, manual polygon classification into drift and fast ice by Muckenhuber et al. (2016) at the cost of efficiency has a potential to better link ice types with environmental factors deemed to control ice conditions, e.g. fast ice with temperature and drift ice with wind and currents. Finally, the binary classification applied here does not allow separating out glacier ice (see Fig. 7)."*

**Line 321. In this study, fast ice was visually identified. There should some criterions to decide which is fast ice, especially where fast ice and pick ice are contiguous.**

Great comment. We have added the information to section 3.3:

*"For the drift ice these were dates of the first and the last presence of ice drifting within the fjord delimited by Worcesterpynten and Palffyodden, excluding short-term episodes in summer and autumn. For the fast ice, the onset (end) was a date when the ice area with clear and stable over the following (preceding) days edges was first (last) visible in the inner parts of the fjord. Notably, freshly formed very thin ice would not be visible."*

**There should be examples of original, segmented and classified SAR images.**

This problem was also raised by Reviewer 1. We have added such a figure (now Fig. 2).